# The First Use of the Washington Group Short Set in a National Survey of Japan: Characteristics of the New Disability Measure in Comparison to an Existing Disability Measure

**DOI:** 10.3390/ijerph21121643

**Published:** 2024-12-10

**Authors:** Takashi Saito, Kumiko Imahashi, Chikako Yamaki

**Affiliations:** 1Department of Social Rehabilitation, Research Institute of National Rehabilitation Center for Persons with Disabilities, 4-1 Namiki, Tokorozawa 359-8555, Japan; imahashi-kumiko@rehab.go.jp; 2Institute for Cancer Control, National Cancer Center, 5-1-1 Tsukiji, Chuou 104-0045, Japan; cyamaki@ncc.go.jp

**Keywords:** Washington Group on Disability Statistics, disability measure, functioning, disability prevalence, Japan

## Abstract

A Japanese national representative survey—the comprehensive survey of living conditions (CSLC)—included the Washington Group Short Set (WGSS) in 2022. This study aimed to characterize the WGSS in comparison to an existing disability measure (EDM), regarding the difference between disability prevalence defined by these two measures and the factors attributable to disagreements between them. A cross-sectional analysis using secondary data from the CSLC (*n* = 32,212) was conducted. The disability prevalences and their ratios (WGSS to EDM) were computed overall and by demographic sub-groups. Binomial logistic regression analyses were performed to explore factors relating to disagreements that functioned to relatively lower/increase the prevalence defined by the WGSS. Consequently, the prevalences defined by the WGSS and EDM were 10.7% (95% confidence interval (CI): 10.4–11.1) and 13.1% (95%CI: 12.7–13.5), respectively. The ratios by the sub-groups were around 0.80–0.90, with the exception of the age-defined sub-group, whose ratios were 0.63 (older sub-group) to 1.23 (child sub-group). Age was the only independent factor associated with two types of disagreements: older age (odds ratios: 1.23–1.80) was associated with disagreement functioning to relatively lower the prevalence defined by the WGSS, and similarly, younger age (ORs: 1.42–2.79) increased the figure. The WGSS may be characterized as being less susceptible to the influence of aging compared to the EDM.

## 1. Introduction

Disability is a global public health issue as it causes disability-based disparities. It can disproportionately compromise people with disabilities (PWD) to access and benefit from heath care services, education, employment, and social services [1]. The definition of disability has changed from a medial model to social model, which conceptualizes disability as arising from the interaction of a person’s functional status with physical, cultural, and policy environments [2]. Moreover, the range of individuals’ conditions which relates to disability has diversified. Specifically, traditional groups of PWD have been considered children with congenital health conditions (e.g., cerebral palsy, birth defects, blindness and deafness) or adults who use a wheelchair, artificial limbs, or assistive devices due to acquired health conditions (e.g., the amputation of extremities and spinal cord injuries) [1,3]. Currently, however, a wider group of individuals who experience difficulties in functioning owing to aging-related health conditions (e.g., noncommunicable diseases, neurological disorders, and chronic diseases) is also recognized as a group of PWD according to the World Health Organization [1]. Considering the fact that aging is a global phenomenon, which is observed not only in developed countries but also in developing countries [4,5], the diversification of PWD’s age groups and their health conditions is seemingly inevitable. The nature of a disability, which is the dynamic interactive phenomenon between individuals and environments, makes it challenging for policy-makers or other stakeholders to measure the disability itself. For tackling this global and evolving challenge, however, robust disability statistics, which use internationally comparable standardized measures, can be a base for assessing the current situation and monitoring the progress of initiatives that ultimately aim to eradicate the disability-based disparity globally.

The Washington Group Short Set (WGSS) is one of the commonly used disability measures, which aims to gather internationally comparable population-based disability statistics. The development of this measure commenced in 2001, led by a United Nations Statistical Commission City Group, the Washington Group on Disability Statistics [6]. The WGSS, developed for the concept of the International Classification of Functioning, Disability, and Health, includes a set of six questions on the following difficulties: seeing, hearing, walking or climbing stairs, remembering or concentrating, self-care, and communication. These simple questions are designed to measure difficulties or functional limitations, rather than restrictions in social participation, with an intention of capturing people at risk of participation restriction in a non-accommodating environment [7]. This feature enables the WGSS to capture disability-based disparities when the measure is included into censuses or surveys in which information on social participation is available and is used to analyze the degree or extent of social participation between those with and without functional limitation based on the social model [7]. As of 2020, over 80 countries, both developing and developed countries, have used the WGSS in censuses or surveys [7,8]. Notably, global agendas, such as the United Nations’ Convention on the Rights of Persons with Disabilities and Sustainable Development Goals, endorse the WGSS as an indicator for assessing the extent of disability-based disparities and ask its parties to use it for reporting their progress towards the specific objectives [7].

In 2022, Japan adapted the WGSS for the first time in an annual national representative survey named the comprehensive survey of living conditions (CSLC). The CSLC includes a variety of questions on living conditions, such as demographic information, health profiles, and socio-economic situations including work, academic attainment, and household income [9]. Combined with these indicators, the WGSS would provide useful information on the current picture of disability-based disparity in Japan. This may serve as a catalyst for better systems and services for approximately 12 million PWD in Japan [10].

As more countries, including globally southern and northern countries, potentially adapt the WGSS as a new disability measure [8], it is important for each country’s stakeholders to understand the implication of its adaptation [11]. Specifically, as different measures of disability measure different things [12], differences occur between disability prevalence defined by the newly adapted WGSS and the existing disability measure (EDM), which has been used as a disability indicator in each country. The differences may cause confusion or difficulties among stakeholders who are involved in policy-making, service planning, or service provision for PWD when they interpret the data. Understanding the implication of the WGSS can contribute to its smooth adaptation and the effective use of the data derived from the WGSS in each country.

In the CSLC, an existing single and comprehensive question on disability has been used since 1989, “Does your health condition affect your daily life?”, to which the answer is “Yes or No”. This question is used by the Japanese government for estimating healthy life expectancy across Japan [13] and monitoring the progress of a nationwide comprehensive health promotion campaign called “Health Japan 21” [14]. Notably, the question serves as an important disability measure in Japan.

Given that the EDM in the CSLC and newly adapted WGSS seemingly captures groups of PWD with different characteristics, disagreements occur between them. The disagreements can be divided into two types: (i) the WGSS showing negative results despite the EDM showing positive results, which relatively lowers the disability prevalence estimations defined by the WGSS and (ii) the WGSS showing positive results while the EDM shows the opposite, which relatively increases the disability prevalence estimations defined by the WGSS. Consequently, these disagreements lead to overall differences in disability prevalence estimations (hereinafter described as prevalence) defined by the two disability measures, which can have a significant impact on policy-making and service provisional plans for PWD in Japan. Therefore, it is important to understand to what extent the prevalence defined by the WGSS differs from that defined by the EDM, and what factors attribute to the disagreements. In particular, knowledge about the magnitude of the effect of aging on the prevalence is crucial from a perspective of Japan as a super-aged society. This knowledge can also be informative for stakeholders in other countries in which rapid aging is occurring or is expected to occur in the near future, because little is known about the magnitude of the effect of aging on WGSS statistics in comparison to other disability measures [11,15,16]. Moreover, single and comprehensive questions, like the EDM in the CSLC, are used not only in Japan but also in other countries including low- and middle-income countries [17]; therefore, knowledge on the difference between the WGSS and EDM in the CSLC can be a reference for these countries’ stakeholders when they newly adapt the WGSS in their national statistics.

This study aimed to understand the characteristics of the WGSS in comparison to the EDM among the Japanese population. First, as a descriptive analysis, the prevalences based on the two disability measures and the agreement/disagreement between the two were calculated. Subsequently, we computed the ratio of the prevalence defined by the WGSS to that defined by the EDM overall and by sub-groups of different age groups and other demographic backgrounds to describe the magnitude of the effect of the demographic factors on the prevalence. Second, we explored the factors of the demographic background and health conditions, which can attribute to disagreements between the two measures, using a multivariate analysis.

For the first research objective, we hypothesized that the prevalence defined by the WGSS would be lower than that of the EDM overall as well as by sub-groups of different demographic backgrounds because of the narrower scope of the WGSS. Theoretically, the EDM, which focuses on the limitation of activities or participation, should estimate a lower disability prevalence than that of the WGSS, which focuses on functional limitations. This is because a portion of individuals experiencing functional limitation should result in the limitation of activities or participation due to interaction with environments. However, the WGSS focuses narrowly on six domains of functional limitation. The EDM, conversely, is not specific; it does not specify types of health conditions and kinds of influence, nor the time frame of the influence due to the health conditions, leaving them up to respondents’ decisions. Therefore, we expected that the prevalence defined by the WGSS would be lower than that defined by the EDM. For the second research objective, owing to the explorational nature, no hypothesis was made.

## 2. Materials and Methods

This was a cross-sectional study using secondary data of the CSLC in 2022. Under article 33 of the Statistical Act [18], the secondary data are provided from the Ministry of Health, Labour, and Welfare of Japan (MHLW) upon requests from researchers or the staff of public organizations who planned to analyze the secondary data for scientific purposes. The authors obtained the secondary data on 5 February 2024. The statistical data shown in the current study are based on the authors’ analysis and may be different from the statistics which were made public by the MHLW.

The current study was approved by the ethics committee of the National Rehabilitation Center for Persons with disabilities (approved No. 2024-069). A procedure acquiring informed consent was waived because the secondary data provided by the MHLW were used in the current study.

### 2.1. Data Source

The CSLC is a nationwide, cross-sectional, and self-administered survey. Since 1986, the CSLC has been conducted annually. In 2022, the survey was conducted from 2nd June to 14th July across Japan.

The questionnaires in the CSLC comprised four modules: household, health, income and savings, and long-term care. This study used data from the household, health, and income and savings modules for analysis. These three modules included data on demographic background, health conditions, and physiological and psychosocial conditions as well as data on the WGSS and EDM.

The questionnaires were distributed to randomly selected citizens across Japan. Specifically, all household members (approximately 674,000 people from 300,000 households) from 5530 randomly selected stratified census tracts received questionnaires and were informed to answer the household and health modules [9]. Similarly, all household members (approximately 70,000 people from 30,000 households) from 2000 census tracts which were randomly selected from the 5530 census tracts were requested to answer the income and savings module [9].

The questionnaires were distributed and collected based on the placement method. Specifically, trained survey staff distributed the questionnaire to each respondent personally. The respondents filled out the answer forms (paper) or typed in their answers (online) by themselves following instructions from the survey staff. Basically, no face-to-face interview was conducted in the CLSC. Then, the respondents’ answers were collected by survey staff personally, returned by mail, or sent through online. All household members, other than those who were under special circumstances (e.g., institutionalization or hospitalization for more than 3 months during the survey period or foreign people who could not understand the Japanese language), were asked to answer the questionnaires. If respondents were not able to answer the questions due to diseases or disabilities, proxies (family members, individuals who provided care, or survey staff) were allowed to answer on their behalf. As the CSLC was designed as a general population survey, no reasonable accommodation for PWD (e.g., a questionnaire with braille for blind people) was provided during the survey. The response rates for each module were as follows: household and health module (68.0%), and income and savings module (61.2%) [9]. Questionnaire samples from the CSLC from 2022 are openly published on the MHLW website (only in Japanese) [19].

### 2.2. Inclusion and Exclusion Criteria for Study Analysis

The data from individuals aged 6 years and over were used for study analysis because those aged 5 years and younger were exempt from answering the WGSS in the CSLC. The exclusion criteria were the data of respondents who provided no answer due to the aforementioned special circumstances or who provided an answer with missing values in variables which were to be used for analysis.

### 2.3. Measures

Variables included for the analysis were the WGSS, the EDM, demographics, health conditions, and physiological and psychosocial variables. Detailed information on these variables, including the original questionnaire (in Japanese), its English version, and each variables’ categorization for analysis are shown in Appendix A.

#### 2.3.1. WGSS and EDM

The WGSS consisted of the following six questions [6]: Do you have difficulty seeing, even if wearing glasses? Do you have difficulty hearing, even if using a hearing aid? Do you have difficulty walking or climbing steps? Do you have difficulty remembering or concentrating? Do you have difficulty (with self-care such as) washing all over or dressing and using your usual language? Do you have difficulty communicating (for example understanding or being understood by others?). The six questions of the WGSS had identical answer options: “no difficulty”, “some difficulty”, “a lot of difficulty”, or “cannot do at all”. Although there are some arguments over how to define disability using the EGSS [20], our study intended to focus on a common way to define disability. Therefore, following the recommendation of the Washington Group on Disability Statistics [6], respondents who answered “a lot of difficulty” or ”cannot to do at all” to any of the six questions were classified as having a disability. Other answer patterns were classified as not having a disability.

Respondents who answered “Yes” to the EDM (“Does your health condition affect your daily life?”) were classified as having a disability and those who answered “No” were classified as not having a disability.

#### 2.3.2. Demographic Variables and Health Conditions

Four types of demographic variables (sex, age, marital status, and living area) were included for this analysis. Age groups were categorized into five groups: 6–19 years (child), 20–39 years (adult), 40–59 years (middle), 60–79 years (older), and 80 years and older (oldest old). Health conditions, specifically the most concerning health conditions which necessitated a respondent to visit hospitals or other health facilities constantly, was defined by two questions on their “constant visit” and “health condition”. Respondents who visited hospitals or other health facilities constantly were asked to choose the most concerning health condition that necessitated them to have constant visits from 42 options of health conditions, including diabetes, stroke, and fracture.

#### 2.3.3. Physiological and Psychosocial Variables

Three types of physiological variables (subjective health status, ethyl alcohol consumption, and smoking habit) and five types of psychosocial variables (educational qualification, subjective financial state, Kessler Psychological Distress Scale [21], health insurance, and employment status) were considered potential confounding factors for disability states and were included in this analysis.

### 2.4. Statistical Analysis

As all variables were categorical, they were presented in numbers and percentages. IBM SPSS (version 28.0.1.0) was used for all statistical analyses. The statistical significance level was set at *p* < 0.05.

First, the prevalence (with 95% confidence intervals (95%CI)) and percentages of agreements/disagreements between the two measures were calculated. Second, the ratio of the prevalence defined by the WGSS to that defined by the EDM was computed overall and by sub-groups with different demographic backgrounds. Finally, we performed two types of binomial logistic regression analyses using the forced entry method (models 1 and 2) to explore the factors contributing to the disagreements between the two measures. The two analyses included only the data of participants aged 20 years and older as some physiological or psychosocial variables (e.g., ethyl alcohol consumption and smoking habit) were limited for participants aged 20 years or older.

In analysis model 1, we explored the factors contributing to the disagreement that functioned to relatively lower the prevalence defined by the WGSS compared with that defined by the EDM; notably, we explored the traits of a portion of individuals who were classified as having a disability by the EDM among those who were classified as having no disability by the WGSS. The disagreement state was used as the objective variable and the demographic background and health conditions were used as the explanatory variables. Conversely, in analysis model 2, we explored the factors attributing to the disagreement that functioned to relatively increase the prevalence defined by the WGSS compared to that defined by the EDM; notably, we explored the traits of a portion of individuals who were classified as having no disability by the EDM among those who were classified as having a disability by the WGSS. The disagreement state was used as the objective variable and the same variables as model 1 were used as the explanatory variables. Only health conditions that were considered the most concerning among 5% or more respondents were used in our two analysis models. The physiological and psychosocial variables were also included in the two analysis models as confounding variables. Correlations between the independent variables (demographic backgrounds, health conditions, and physiological and psychosocial variables) were not considered high, as the phi coefficient and Cramer’s coefficient of association between them were 0.007–0.610.

## 3. Results

The data selection process is shown in Figure 1. From the original data of 45,160 study participants who responded to the household, health, and income and savings modules, we omitted the data that met the exclusion criteria or those with missing values. Consequently, the number of eligible data for analysis was 32,212. Information on the number of study participants with missing values for each variable is available in Appendix A.

Characteristics of the data eligible for analysis is presented in Table 1. The prevalences defined by the WGSS and EDM were 10.7% (95%CI; 10.4–11.1) and 13.1% (95%CI; 12.7–13.5), respectively. The prevalence of each of the six questions of the WGSS was 3.2–6.3% (the lowest percentage was difficulty for self-care and communication and the highest percentage was difficulty for mobility).

The agreement/disagreement between the two measures are illustrated schematically in Figure 2. Overall, nearly 88% of respondents provided similar responses across the two disability measures (Figure 2(A,D)). Meanwhile, approximately 12% of respondents showed disagreements in their responses (Figure 2(B,C)). Of the 28,759 respondents who were defined as having no disability by the WGSS (Figure 2(C,D)), 2378 respondents (8.27%, Figure 2(C)) were defined as having a disability by the EDM. Of the 3453 respondents who were defined as having a disability by the WGSS (Figure 2(A,B)), 1613 respondents (46.7%, Figure 2(B)) were defined as having no disability by the EDM.

Table 2 shows the prevalence defined by the two disability measures and their ratios overall and by the sub-groups. The WGSS-defined prevalence was relatively lower than that defined by the EDM; the ratio was 0.82 overall. Across the sub-groups, most of the ratios were around 0.80–0.90. One exception, however, was the ratio in the sub-group defined by age. The ratio fluctuated widely and ranged from 0.63 to 1.23. Lower ratios, indicating a relatively lower prevalence defined by the WGSS, were observed for the middle (0.74, aged 40–59) and older (0.63, aged 60–79) age groups. A higher ratio, indicating a relatively increased prevalence defined by the WGSS, was observed for the child (1.23, aged 6–19) age group.

The trajectory of the prevalence defined by the WGSS along the age groups showed that the figure remained stable at around 5–6% for the child to middle age groups (Table 2). The figure increased to around 10% for the older age group, being two times higher compared to the child age group. The ratio finally reached around 40% for the oldest old age group. Meanwhile, the figure defined by the EDM showed a different trajectory, especially for the middle to older age groups. The figure started from 5 to 6% among the child to adult age groups, being almost similar to its counterparts. However, the figure showed a relatively rapid increase in the middle to older age groups, being almost 15% for the older age group, three times higher than that of the child age group. Finally, the figure reached almost the same percentage as its counterpart, around 40%, for the oldest old age group (80 years and older).

The results of the binominal logistic regression are shown in Table 3 and Table 4. Table 3 shows that age (older, odds ratios (ORs: 1.23–1.80), marital status (divorced or widowed, OR: 1.27), three kinds of health conditions (low back pain (OR:1.76), depression and other mental conditions (OR: 1.66), and arthritis (OR:2.68)) had significant and positive associations with the disagreement, which functioned to relatively lower the prevalence defined by the WGSS in comparison to that defined by the EDM. Table 4 showed that age (younger, ORs: 1.42–2.79) and two kinds of health conditions (hypertension (OR:1.63) and diabetes (OR:2.25)) had significant and positive associations with the disagreement, which functioned to relatively increase the prevalence defined by the WGSS. The detailed information of the binominal logistic regression analysis, which additionally showed the results of confounding factors (physiological variables and socio-economic variables), are available in Appendix A.

## 4. Discussion

### 4.1. Key Points

The purpose of this current study was to understand the characteristics of the WGSS in comparison to the EDM among the Japanese population. Our results revealed a relatively lower prevalence defined by the WGSS in comparison to that of the EDM. Moreover, age was the only factor that was independently associated with the two types of disagreements between the two measures; older age (ORs: 1.23–1.80) was associated with the disagreement that relatively lowered the prevalence defined by the WGSS, and younger age (ORs: 1.42–2.79) was associated with another disagreement that relatively increased the figure. These findings implied that the WGSS may be characterized as a measure less susceptible to the influence of aging compared to the EDM. Owing to the high percentage of the older Japanese population (30% as of 2022 [22]), the overall prevalence defined by the WGSS, consequently, became lower in comparison to the figure defined by the EDM. This knowledge needs to be considered when interpreting the disability statistics based on the WGSS and while utilizing them.

### 4.2. Age and Differences Between the Two Disability Prevalence Estimations

Supporting our hypothesis, our descriptive analysis showed that the prevalence defined by the WGSS was lower compared to that of the EDM, especially in the middle to older age groups. This observed relationship was consistent with the results of the multivariate analysis, showcasing that older age was independently and significantly associated with one of the two types of disagreement that functioned to relatively lower the disability prevalence defined by the WGSS. This was the most notable finding in our study.

Our descriptive analysis showed that the disability prevalence based on the WGSS started from around 5–6% for the child to adult age groups and eventually increased to 10% for the older age group. This data trend is seemingly compatible with those in other countries. Weeks et al. [15] reported that disability estimates based on the WGSS in the U.S. were 3.2% in the 18–44 years group, 11.6% in the 45–64 years group, and 19.8% in the 65 years and older group. Moreover, the average disability prevalences defined by the WGSS in 21 low- and middle-income countries, which were published as supplementary data in a study report by Mitra and Yap [23], were 2.0% in the 30–44 years group, 5.6% in the 45–64 years group, and 17.5% in the 65 years and older group (these figures were calculated by the authors based on the supplementary data). Admittedly, direct comparison between the current data and these previously reported data from other countries is impossible due to differences in the prerequisite conditions across the studies. The survey in the U.S. [15] used a face-to-face interview method, whereas self-administered surveys were conducted in the CSLC. These data, however, showed a similarity in their overall data trends, i.e., the figure started from approximately 5% for the child to adult age groups and increased to approximately 10% for the middle or older age groups. Our disability prevalence estimate defined by the WGSS was unlikely to be excessively deviated (lower) from the data trends of other countries. Therefore, the relatively lower prevalence defined by the WGSS was seemingly attributed to the data trend of the EDM-defined prevalence, which showcased a more rapid increase than that of the WGSS among the age groups.

As we hypothesized, the WGSS estimated a lower disability prevalence than the EDM. This phenomenon was obviously observed among the middle to older age groups. This might be attributed to the fact that the EDM could capture a wider group of individuals whose daily lives were influenced by aging-related health issues. However, the remarkable differences between the two disability measures disappeared among the oldest older age group, 80 years and older. The prevailing aging-related functional limitations such as frailty and cognitive decline [24] and limited activity or social participation, such as impaired basic or instrumental activity of daily living [25,26,27], among the oldest older age group may have led the prevalences defined by the two measures to rapidly increase, resulting in equivalent figures between them. This result implied that differences for estimating the prevalences between the two disability measures could be ignorable when the two measures were used among the oldest older generation.

Interestingly, our descriptive analysis showed that the prevalence defined by the WGSS was higher than that defined by the EDM only among the child age group (6–19 years). Our multivariate analysis also showed a significant and positive association between younger age and the disagreement between two measures which functioned to relatively increase the prevalence defined by the WGSS. Basically, Japanese children and adolescents are protected by their parent(s). Even if some of them have functional limitations induced by health conditions, their daily lives are supported by their parent(s), school teachers, social welfare services, or informal supporters in their communities. Of course, these formal/informal supports for Japanese children with a disability may not necessarily be adequate. However, many of them may consider themselves not being affected in their daily life due to health conditions or functional limitations. This may have led to the relatively higher prevalence defined by the WGSS compared to that defined by the EDM among the children.

### 4.3. Other Factors Associated with the Disagreement That Functioned to Relatively Lower the Disability Prevalence Estimates Defined by the WGSS

Our multivariate analysis showed that marital state, some health conditions, and older age were independently associated with the disagreement that functioned to relatively decrease the prevalence defined by the WGSS. Specifically, individuals who were classified as having a disability by the EDM, despite having no disability by the WGSS, had the demographic traits of being divorced or widowed or had the health conditions of low back pain, depression and other mental conditions, or arthritis.

Marital status is a determinant of health, especially among middle to older generations [28,29,30]. Bennett [28], based on a longitudinal study, reported a wider influence of marital status, especially widowed and divorced statuses, on health, welfare service use, self-rated health, and activity limitation among generations of 40 years old and older. The wider influence of marital status might be more easily reflected by the EDM than by the WGSS due to its wider scope. This may be a possible reason for the relationship we found between marital state (divorced or widowed) and the disagreement.

Depression and other mental conditions were associated with the disagreement. This finding was consistent with the WGSS’s limitations, which have been well documented; the WGSS is not able to record individuals with psychosocial disabilities due to the lack of questions regarding mental issues in the WGSS [6]. The relatively lower prevalence of the WGSS was, in part, due to the underestimation of individuals with mental health issues.

Low back pain and arthritis, commonly categorized as musculoskeletal conditions, were the contributing factors to the disagreement, which functioned to lower the prevalence defined by the WGSS. These two musculoskeletal conditions can seemingly cause mobility and/or self-care issues, which are reflected by the sub-questions of the WGSS (walking or climbing stairs or self-care). These sub-questions, however, focus only on lower-level functioning [31], which is potentially compromised among people who have severe musculoskeletal conditions only. Conversely, issues relating to higher-level functioning, which may involve activities relevant to instrumental activities of daily living or working, may be widely reflected by the EDM among those who have low back pain or arthritis with mild to moderate symptoms as well as severe symptoms. Consequently, the WGSS’s limited scope on lower-level functioning might have resulted in its relatively lower prevalence estimation in comparison to the EDM. This is a possible explanation for our finding that low back pain and arthritis showed an association with the disagreement which lowered the prevalence defined by the WGSS. Notably, low back pain [32] and knee arthritis [33], a type of arthritis, have been reported as leading causes of disabilities globally. In Japan, a question on self-reported symptoms in the CSLC in 2022, which was not analyzed in the current study, showed that low back pain and shoulder pain were the most common symptoms in both men (91.6 people/1000 people, 53.3 people/1000 people) and women (111.9 people/1000 people, 105.4 people/1000 people), respectively [9]. Given the high prevalence of musculoskeletal conditions, the possible underestimation that may be relevant to low back pain or arthritis need to be considered when the WGSS is used.

### 4.4. Other Factors Associated with the Disagreement That Relatively Increased the Disability Prevalence Estimates Defined by the WGSS

Our multivariate analysis showed that two health conditions and younger age were independently associated with the disagreement, which functioned to increase the prevalence defined by the WGSS. Specifically, individuals who were classified as having no disability by the EDM, despite being defined as having a disability by the WDSS, had the health conditions of hypertension or diabetes.

Hypertension and diabetes are well-known risk factors for some conditions, which result in severe functional limitations and/or limited social participation, such as those due to stroke or heart diseases [34,35]. However, the two health conditions rarely induce serious functional difficulties by themselves. Some people with hypertension or diabetes can even be asymptomatic. Thus, the exact reason for the association of hypertension and diabetes with the disagreement was unclear. One possible explanation, however, is the higher prevalence of lifestyle diseases, including hypertension and diabetes, among PWD than individuals without disabilities [36,37,38]. Cragg et al. [37] analyzed data from Canada’s cross-sectional representative data (n = 60,678) and revealed that spinal cord injury was strongly associated with type 2 diabetes after controlling confounding factors. Some PWD who have functional difficulties but who can cope with their daily lives with formal/informal supports may answer as “having difficulty” in the WGSS but “no influence (no disability)” in the EDM. If some of these PWD have hypertension or diabetes, they may be concerned about these secondary conditions, rather than their primary condition, because primary conditions, in many cases, cannot change or cannot be curable, but their secondary conditions can be controllable depending on their lifestyle modifications or medications. This might be a possible explanation for our finding that hypertension and diabetes were associated with the disagreement.

### 4.5. Implication for Use of the WGSS

Our study showed that the WGSS in the CSLC provided an overall lower prevalence than the EDM. This finding was mainly attributed to the WGSS’s characteristics of being narrower in scope and less susceptible to the influence of aging in comparison to the EDM and overall aging in Japan. If the Japanese government substituted the WGSS for the EDM for estimating the prevalence, the figure would show an ostensible decline. This apparent decline in the prevalence may lead to the underestimation of overall demand for services for PWD or execrated effect estimations of initiatives or campaigns for PWD. This may be a drawback in using the WGSS as a substitution for the EDM.

Meanwhile, of the 3453 respondents categorized as “having disability” by the WGSS, approximately half of them were categorized as “no disability” by the EDM. As mentioned above, some PWD with certain functional difficulties but who are able to cope with their daily lives with formal/informal supports may not be aware of the disabilities’ influence on their lives. This could lead to the potential underestimation of the service needs among PWD with functional limitations but who are not aware of the influence relating to their conditions. Hall et al. [39] argued that, even if PWD do not consider themselves limited, they should be recognized as people in need for PWD-related services as far as they have some difficulties due to health conditions or disabilities that affect their daily life. Use of the WGSS may be an opportunity to unveil neglected potential needs that remain unrecognized when the EDM is used. This may be an advantage of using the WGSS as a substitute for the EDM.

### 4.6. Limitations

This study has some limitations. First, the CSLC, in which no reasonable accommodation for PWD was provided, may exclude potential respondents with disabilities. This may have underestimated the prevalence defined by the WGSS and EDM. Further, this may have skewed the relationship between them, owing to selection bias. Second, considering the self-reporting nature of CSLC, responses about health conditions might have included inaccurate information, which might have interfered with our study results. Third, no data on disability certificates were available in the CSLC. For better understanding of the characteristics of the WGSS, it is crucial to know the extent of the possession of disability certificates reflected by the WGSS. Further studies are needed to examine the relationship. Finally, other thresholds in the WGSS (e.g., considering “some difficulty” disability) would have shown a different prevalence and a different relationship with the EDM. As there is no gold standard for measuring “true disability”, the influence of other thresholds in the WGSS need to be assessed in further studies.

## 5. Conclusions

The WGSS in the CSLC had the characteristics of being less susceptible to the influence of aging compared to the EDM due to its narrower scope. This resulted in an overall lower prevalence defined by the WGSS compared to that defined by the EDM, owing to the high percentage of the elderly Japanese population. This knowledge needs to be considered when stakeholders interpret disability statistics based on the WGSS and utilize them.

## Figures and Tables

**Figure 1 ijerph-21-01643-f001:**
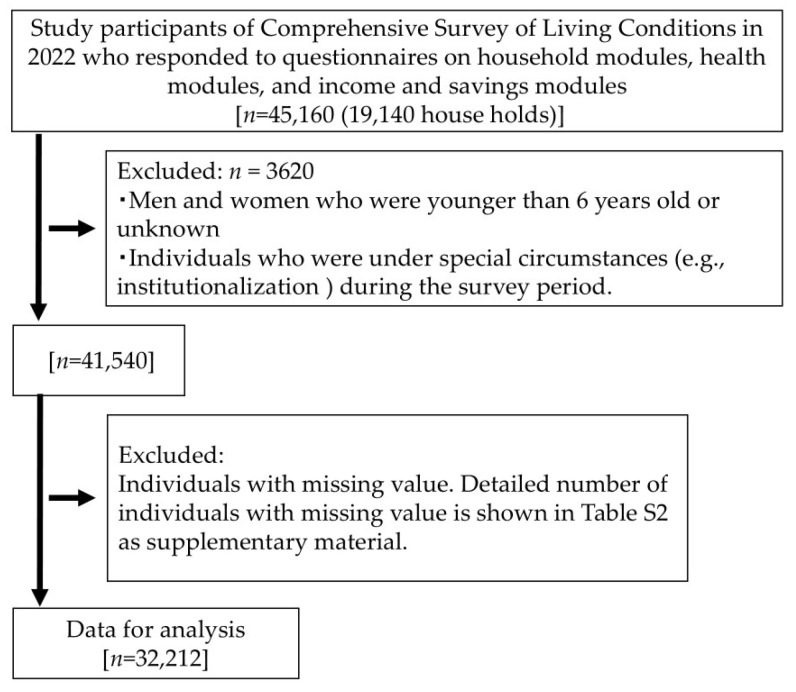
Participant selection process.

**Figure 2 ijerph-21-01643-f002:**
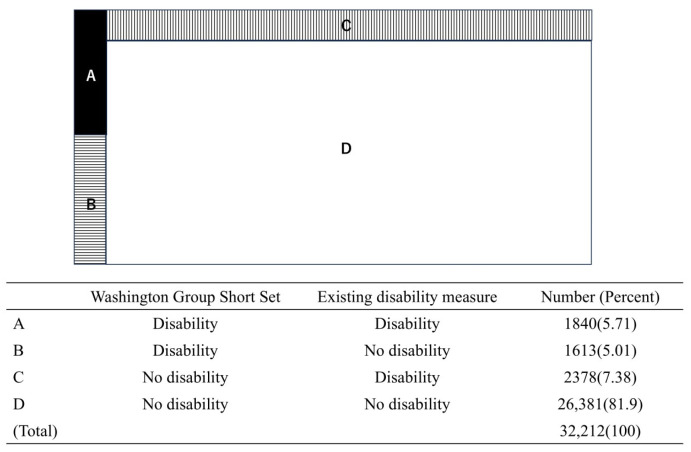
Agreement and disagreement between the two disability measures. The are of A, B, C and D schematically illustrated the number of individuals who were categorized as disability or no-disability by the Washington Group Short Set and the existing disability measure. The Washington Group Short Set included a set of six questions on the following difficulties: seeing, hearing, walking or climbing stairs, remembering or concentrating, self-care, and communication. Answer options were “no difficulty”, “some difficulty”, “a lot of difficulty”, or “cannot do at all”. Respondents who answered “a lot of difficulty” or ”cannot to do at all” to any of the six questions were classified as having a disability. Other answer patterns were classified as having no disability. For the existing disability measure, “Does your health condition affect your daily life?”, the answer option was “Yes or No”. Respondents who answered “Yes” were classified as having a disability and those who answered “No” were defined as having no disability.

**Table 1 ijerph-21-01643-t001:** Characteristics of study participants (n = 32,212).

	Number	Percent (95% Confidence Interval)	
Sex			
Male	15,368	47.7	(47.2–48.3)
Female	16,844	52.3	(51.7–52.8)
Age (years)			
Child, 6–19	4205	13.1	(12.7–13.4)
Adult, 20–39	5552	17.2	(16.8–17.7)
Middle, 40–59	9310	28.9	(28.4–29.4)
Older, 60–79	10,237	31.8	(31.3–32.3)
Oldest old, 80 and over	2908	9.0	(8.7–9.3)
Marital status			
Married	18,232	56.6	(56.1–57.1)
Single	9895	30.7	(30.2–31.2)
Divorced/widowed	4085	12.7	(12.3–13.0)
Living area			
Area with 150,000 or more inhabitants	15,370	47.7	(47.2–48.3)
Area with less than 150,000 inhabitants	16,842	52.3	(51.7–52.8)
Constant visits to hospitals			
Yes (constant visits)	15,012	46.6	(46.1–47.1)
No (non-constant visits)	17,200	53.4	(52.9–53.9)
Subjective health status			
Good	13,500	41.9	(41.4–42.4)
Normal	14,942	46.4	(45.8–46.9)
Bad	3770	11.7	(11.4–12.1)
Ethyl alcohol consumption ^(b)^			
Never or quit drink	16,468	58.8	(58.2–59.4)
Social drinker or low-risk group (>0 to ≤100 g/week)	6061	21.6	(21.2–22.1)
Middle-risk group (>100 to ≤350 g/week)	4668	16.7	(16.2–17.1)
High-risk group (>350 g/week)	810	2.9	(2.7–3.1)
Smoking habit ^(b)^			
Never/ex-smoker	23,707	84.6	(84.2–85.1)
Current smoker	4300	15.4	(14.9–15.8)
Educational qualification ^(a)^			
Vocational school/junior college/community (technical) college/university/postgraduate school	13,272	45.0	(44.4–45.5)
High school	13,027	44.1	(43.6–44.7)
Primary/junior high school	3221	10.9	(10.6–11.3)
Subjective financial state			
Wealthy	1933	6.0	(5.7–6.3)
Not poor nor wealthy	13,338	41.4	(40.9–41.9)
Poor	16,941	52.6	(52.0–53.1)
Kessler Psychological Distress Scale ^(c)^			
Normal (total score ≤ 4)	23,017	75.5	(75.1–76.0)
Mild illness (5 ≤ total score ≤ 12)	6271	20.6	(20.1–21.0)
Severe illness (13 ≤ total score)	1182	3.9	(3.7–4.1)
Health insurance			
Employee insurance	19,463	60.4	(59.9–61.0)
National Health Insurance	7468	23.2	(22.7–23.6)
Other	5281	16.4	(16.0–16.8)
Employment status ^(a)^			
Employed	14,464	49.0	(48.4–49.6)
Self-employed	1815	6.1	(5.9–6.4)
Employed (other)	1801	6.1	(5.8–6.4)
Unemployed	11,440	38.8	(38.2–39.3)
Disability defined by existing measure of disability ^(d)^			
Yes (disability)	4218	13.1	(12.7–13.5)
No (no disability)	27,994	86.9	(86.5–87.3)
Disability defined by WGSS			
No (no disability)	28,759	89.3	(88.9–89.6)
Yes (disability)	3453	10.7	(10.4–11.1)
WGSS, vision			
No (no disability)	30,986	96.2	(96.0–96.4)
Yes (disability)	1226	3.8	(3.6–4.0)
WGSS, hearing			
No (no disability)	31,085	96.5	(96.3–96.7)
Yes (disability)	1127	3.5	(3.3–3.7)
WGSS, mobility			
No (no disability)	30,171	93.7	(93.4–93.9)
Yes (disability)	2041	6.3	(6.1–6.6)
WGSS, cognition			
No (no disability)	30,924	96.0	(95.8–96.2)
Yes (disability)	1288	4.0	(3.8–4.2)
WGSS, self-care			
No (no disability)	31,185	96.8	(96.6–97.0)
Yes (disability)	1027	3.2	(3.0–3.4)
WGSS, communication			
No (no disability)	31,191	96.8	(96.6–97.0)
Yes (disability)	1021	3.2	(3.0–3.4)

^(a)^ Only participants who were 15 years and over (n = 29,520). ^(b)^ Only participants who were 20 years and over (n = 28,007). ^(c)^ Only participants who were 12 years and over (n = 30,470). ^(d)^ Specific question: Does your health condition affect your daily life? WGSS: Washington Group Short Set.

**Table 2 ijerph-21-01643-t002:** The number and percentages of study participants with disabilities defined by the WGSS and the existing measure of disability according to demographic backgrounds.

	WGSS Disability Measure (A)			Existing Measure of Disability ^(a)^ (B)			A:B Ratio
	Number	Percent (95% Confidence Interval)		Number	Percent (95% Confidence Interval)		
Total (n = 32,212)	3453	10.7	(10.4–11.1)	4218	13.1	(12.7–13.5)	0.82
Sex							
Male (n = 15,368)	1498	9.75	(9.3–10.2)	1804	11.7	(11.2–12.3)	0.83
Female (n = 16,844)	1955	11.6	(11.1–12.1)	2414	14.3	(13.8–14.9)	0.81
Age (years)							
Child, 6–19 (n = 4205)	229	5.4	(4.8–6.2)	186	4.4	(3.8–5.1)	1.23
Adult, 20–39 (n = 5552)	329	5.9	(5.3–6.6)	342	6.2	(5.6–6.8)	0.96
Middle, 40–59 (n = 9310)	577	6.2	(5.7–6.7)	783	8.4	(7.9–9.0)	0.74
Older, 60–79 (n = 10,237)	1073	10.5	(9.9–1.1)	1716	16.8	(16.0–17.5)	0.63
Oldest old, 80 and over (n = 2908)	1245	42.8	(41.0–44.6)	1191	41.0	(39.2–42.8)	1.05
Marital status							
Married (n = 18,232)	1767	9.7	(9.3–10.1)	2347	12.9	(12.4–13.4)	0.75
Single (n = 9895)	668	6.8	(6.3–7.3)	764	7.7	(7.2–8.3)	0.87
Divorced/widowed (n = 4085)	1018	24.9	(23.6–26.3)	1107	27.1	(25.8–28.5)	0.92
Living area							
Area with 150,000 or more inhabitants (n = 15,370)	1579	10.3	(9.8–10.8)	1966	12.8	(12.3–13.3)	0.80
Area with less than 150,000 inhabitants (n = 16,842)	1874	11.1	(10.7–11.6)	2252	13.4	(12.9–13.9)	0.83

^(a)^ Specific question: Does your health condition affect your daily life? WGSS: Washington Group Short Set.

**Table 3 ijerph-21-01643-t003:** Chi square test and binomial logistic regression analysis for the disagreement between the two disability measures that relatively lowered the disability prevalence estimates defined by the WGSS compared to those defined by the existing disability measure (participants aged 20 years and over).

	Disability				Chi Square Test	Binomial Logistic Regression ^(d)^	
	WGSS: No Existing Measure: No (n = 22,551)		WGSS: No Existing Measure: Yes (n = 2232)				
	Number	(Percent)	Number	(Percent)	*p*-Value	Adjusted Odds Ratio	(95% Confidence Interval)
Sex					0.001		
Male	10,894 ^†^	(48.31)	999	(44.76)		Reference	
Female	11,657 ^†^	(51.69)	1233	(55.24)		0.90	(0.80–1.01)
Age (years)					<0.001		
20–39	4979 ^†^	(22.08)	244	(10.93)		Reference	
40–59	8155 ^†^	(36.16)	578	(25.90)		1.23	(1.01–1.48)
60–79	8060 ^†^	(35.74)	1104	(49.46)		1.51	(1.21–1.89)
80 and over	1357 ^†^	(6.02)	306	(13.71)		1.80	(1.32–2.44)
Marital status					<0.001		
Married	15,061 ^†^	(66.79)	1404	(62.90)		Reference	
Single	4862 ^†^	(21.56)	389	(17.43)		1.13	(0.96–1.32)
Divorced/widowed	2628 ^†^	(11.65)	439	(19.67)		1.27	(1.09–1.47)
Living area					0.97		
Area with 150,000 or more inhabitants	10,812	(47.94)	1071	(47.98)		Reference	
Area with less than 150,000 inhabitants	11,739	(52.06)	1161	(52.02)		1.03	(0.93–1.14)
The most concerning health condition ^(a)^: low back pain					<0.001		
No ^(b)^	22,102 ^†^	(98.01)	2022	(90.59)		Reference	
Yes ^(c)^	449 ^†^	(1.99)	210	(9.41)		1.76	(1.42–2.17)
The most concerning condition ^(a)^: depression and other mental conditions					<0.001		
No ^(b)^	22,316 ^†^	(98.96)	2059	(92.25)		Reference	
Yes ^(c)^	235 ^†^	(1.04)	173	(7.75)		1.66	(1.28–2.15)
The most concerning health condition ^(a)^: hypertension					<0.001		
No ^(b)^	20,286 ^†^	(89.96)	2087	(93.50)		Reference	
Yes ^(c)^	2265 ^†^	(10.04)	145	(6.50)		0.33	(0.27–0.40)
The most concerning health condition ^(a)^: diabetes					0.003		
No ^(b)^	21,624 ^†^	(95.89)	2111	(94.58)		Reference	
Yes ^(c)^	927 ^†^	(4.11)	121	(5.42)		0.56	(0.44–0.70)
The most concerning health condition ^(a)^: arthritis					<0.001		
No ^(b)^	22,363 ^†^	(99.17)	2116	(94.80)		Reference	
Yes ^(c)^	188 ^†^	(0.83)	116	(5.20)		2.68	(2.02–3.55)
The most concerning health condition ^(a)^: dyslipidemia					0.002		
No ^(b)^	21,852 ^†^	(96.90)	2189	(98.07)		Reference	
Yes ^(c)^	699 ^†^	(3.10)	43	(1.93)		0.38	(0.27–0.53)
The most concerning health condition ^(a)^: dental diseases					0.02		
No ^(b)^	21,923 ^†^	(97.22)	2188	(98.03)		Reference	
Yes ^(c)^	628 ^†^	(2.78)	44	(1.97)		0.43	(0.30–0.60)
The most concerning health condition ^(a)^: eye diseases					0.39		
No ^(b)^	22,030	(97.69)	2174	(97.40)		Reference	
Yes ^(c)^	521	(2.31)	58	(2.60)		0.47	(0.34–0.64)

^(a)^ The condition which necessitated a study participant to visit hospital constantly and was also considered the most concerning condition among his/her conditions. ^(b)^ “No” includes study participants who did not visit hospital constantly or who visited hospital constantly but did not have the specific condition nor consider the specific condition as the most concerning. ^(c)^ “Yes” includes study participants who visited hospital constantly due to the specific condition that was considered the most concerning condition. Only the health conditions that were considered the most concerning condition among at least 5% of study participants who constantly visited hospital in either group (existing measure—no and WGSS—no (n = 9879) and existing measure—yes and WGSS—no (n = 1908)) were incorporated in the multivariate logistic regression analysis. ^(d)^ Physiological (subjective health status, ethyl alcohol consumption, and smoking habit) and socio-economic variables (educational qualification, subjective financial state, Kessler Psychological Distress Scale, health insurance, and employment status) were incorporated in the analysis model using the forced entry method. Detailed results of the analysis are available in Appendix A. ^†^ The adjusted residual values exceeded 1.96 or were below −1.96.

**Table 4 ijerph-21-01643-t004:** Chi square test and binomial logistic regression analysis for the disagreement between the two disability measures that relatively increased the disability prevalence estimates defined by the WGSS compared to those defined by the existing disability measure (participants aged 20 years and over).

	Disability				Chi Square Test	Binomial Logistic Regression ^(d)^	
	WGSS: Yes Existing Measure: Yes (n = 1800)		WGSS: Yes Existing Measure: No (n = 1424)				
	Number	(Percent)	Number	(Percent)	*p*-Value	Adjusted Odds Ratio	(95% Confidence Interval)
Sex					<0.001		
Male	697 ^†^	(38.72)	680	(47.75)		Reference	
Female	1103 ^†^	(61.28)	744	(52.25)		0.96	(0.77–1.18)
Age					<0.001		
80 and over	885 ^†^	(49.17)	360	(25.28)		Reference	
60–79	612	(34.00)	461	(32.37)		1.42	(1.04–1.92)
40–59	205 ^†^	(11.39)	372	(26.12)		1.87	(1.19–2.93)
20–39	98 ^†^	(5.44)	231	(16.22)		2.79	(1.60–4.84)
Marital status					<0.001		
Married	943 ^†^	(52.39)	824	(57.87)		Reference	
Single	189 ^†^	(10.50)	250	(17.56)		0.71	(0.50–0.99)
Divorced/widowed	668 ^†^	(37.11)	350	(24.58)		0.95	(0.75–1.20)
Living area					0.88		
Area with 150,000 or more inhabitants	817	(45.39)	650	(45.65)		Reference	
Area with less than 150,000 inhabitants	983	(54.61)	774	(54.35)		1.16	(0.97–1.39)
The most concerning health condition ^(a)^: hypertension					0.01		
No ^(b)^	1615 ^†^	(89.72)	1236	(86.80)		Reference	
Yes ^(c)^	185 ^†^	(10.28)	188	(13.20)		1.63	(1.23–2.17)
The most concerning health condition ^(a)^: low back pain					<0.001		
No ^(b)^	1647 ^†^	(91.50)	1366	(95.93)		Reference	
Yes ^(c)^	153 ^†^	(8.50)	58	(4.07)		0.90	(0.62–1.32)
The most concerning health condition ^(a)^: depression and other mental conditions					<0.001		
No ^(b)^	1690 ^†^	(93.89)	1400	(98.31)		Reference	
Yes ^(c)^	110 ^†^	(6.11)	24	(1.69)		0.46	(0.27–0.81)
The most concerning health condition ^(a)^: dementia					<0.001		
No ^(b)^	1696 ^†^	(94.22)	1406	(98.74)		Reference	
Yes ^(c)^	104 ^†^	(5.78)	18	(1.26)		0.36	(0.21–0.64)
The most concerning health condition ^(a)^: arthritis					<0.001		
No ^(b)^	1697 ^†^	(94.28)	1401	(98.38)		Reference	
Yes ^(c)^	103 ^†^	(5.72)	23	(1.62)		0.46	(0.28–0.78)
The most concerning health condition ^(a)^: eye diseases					0.02		
No ^(b)^	1703 ^†^	(94.61)	1373	(96.42)		Reference	
Yes ^(c)^	97 ^†^	(5.39)	51	(3.58)		1.03	(0.67–1.57)
The most concerning health condition ^(a)^: diabetes					0.06		
No ^(b)^	1709	(94.94)	1330	(93.40)		Reference	
Yes ^(c)^	91	(5.06)	94	(6.60)		2.25	(1.56–3.25)

^(a)^ The condition which necessitated a study participant to have constant hospital visits and was also considered the most concerning condition among his/her conditions. ^(b)^ “No” includes study participants who did not visit hospital constantly or who visited hospital constantly but did not have the specific condition nor consider the specific condition as the most concerning. ^(c)^ “Yes” includes study participants who visited hospital constantly due to the specific condition that was considered the most concerning condition. Only the health conditions that were considered the most concerning condition among at least 5% of study participants who visited hospital constantly in either group (existing measure—yes and WGSS—yes (n = 1645); existing measure—no and WGSS—yes (n = 869)) were incorporated in the multivariate logistic regression analysis. ^(d)^ Physiological (subjective health status, ethyl alcohol consumptions, and smoking habit) and socio-economic variables (educational qualification, subjective financial state, Kessler Psychological Distress Scale, health insurance, and employment status) were incorporated in the analysis model using the forced entry method. Detailed results of the analysis are available in Appendix A. ^†^ The adjusted residual values exceeded 1.96 or were below −1.96.

## Data Availability

No new data were created or analyzed in this study. Data sharing is not applicable to this study.

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
