# Peer review of "The First Use of the Washington Group Short Set in a National Survey of Japan: Characteristics of the New Disability Measure in Comparison to an Existing Disability Measure"

_ijerph, 2024, doi:10.3390/ijerph21121643_

Round 1
Reviewer 1 Report
Comments and Suggestions for Authors
The methodology and results of comparing prevalence rates using the WG-SS and EDM are clear and well-presented. However, I do have an issue with the discussion and interpretation. I think these are important points and if addressed in the paper would make it a greater contribution.
People are often concerned when disability prevalence measures don't align perfectly. In fact, the WG-SS and EDM actually align pretty well, but the EDM is slightly higher. But -- and this is key -- they are not measuring the same thing. Disability is in fact not truly a binary variable. Difficulty in functioning exists along a continuum and so prevalence is dependent on where one establishes the threshold. The EDM is drawing the threshold at ANY influence on daily life. The WG-SS is drawing the line at "a lot of difficulty" or "cannot do." My guess is, that if the researchers included those answering some difficulty to multiple questions, that the WG-SS would identify many more people than the EDM. And if the threshold was "some" to only one question, it would be way, way higher than the EDM.
So which is the right threshold? The WG recommended their threshold for two reasons. (1) in international testing, that threshold turned out to be more internationally comparable. (2) They wanted to identify a population that clearly was at higher risk of exclusion. When we look at outcome indicators for people with only "some" difficulty and even those with "some" difficulty in multiple functional domains we see lower levels of exclusion.
So who are the people we are trying to identify? Those clearly at risk of exclusion, or those whose lives are affected in any way? There is no right answer here. It depends on your purpose. But it isn't really fair to say the WG-SS is misidentifying people who the EDM captures. What if those that are identified by the EDM but not the WG-SS reported that they had some difficulty doing things? Then both the EDM and the WG-SS are picking up the impact of their situation.
It would be interesting to see the relation between the EDM and the WG-SS using different thresholds. THe typical ones are:
a) a lot or cannot do
b) some in multiple domains
c) some in a single domain
I believe the paper would be better served by showing the relationship between EDM and both (a) and (b). And to include in the discussion the idea of a threshold and what threshold is appropriate for what purpose - and that in comparing prevalences, that threshold is very important to keep in mind.
The whole "threshold" idea is very important. Otherwise people have the mistaken belief that there is one gold standard measure of "true" disability.
Reviewer 2 Report
Comments and Suggestions for Authors
Dear authors,
Thank you for allowing me to read your paper. The question of comparing the Washington Group Short Set and the existing instrument within a Japanese national survey is important to understand differences in disability prevalence. The article has important practical implications, and it could be of great interest for readers. The manuscript is well organized, well-written, and has a good flow. I consider it worth publishing, and I have only minor suggestions:
- Check the orthography of the title, also it seems long, if you could make it shorter.
- The abstract is well written, it provides information on the aim of the study, the methodology and some main findings are discussed.
- The introduction is well written, the relevance of the study is well justified, it was identified the research gap.
- You present mostly the situation in Japan; it would be interesting to see how it fits in other countries (Asian or European context?). At some point you write “As more countries or national surveys potentially adapt the WGSS as a new disability measure, it is important for each countries’ stakeholders to understand the implication of its adaptation…” If you could expand and provide some examples.
- The aim and the research hypothesis are clear.
- The methodology is detailed and well explained, the instruments used are presented with enough information.
- The results are clearly presented, the figures and tables are formatted accordingly. The only aspect is that they are quite long, you present all the data from your questionnaire, maybe to put some of it in supplementary material, it is a suggestion, if you decide to keep it in the manuscript, it could work as well.
- Regarding the discussion section, you could mention the study aim.
- Also, if you could expand conclusion section and mention also the practical implications and some future research directions.
Best regards
Round 2
Reviewer 1 Report
Comments and Suggestions for Authors
1. The introduction needs a better explanation of the switch over time from a medical/impairment based way of defining disability to a functional approach, and how that relates to the growing use of the social model of disability, as embodied in the Convention on the Rights of Persons with Disabilities. A place to start is: https://documents.worldbank.org/en/publication/documents-reports/documentdetail/578731468323969519/measuring-disability-prevalence
2. The EDM is a yes/no question and the WG-SS has scaled responses. It would be very interesting for the authors to take advantage of this fact. How many of those responding "cannot do" to a WG-SS question were identified by the EDM? How many who only are identified as having a disability with the "lot of difficulty" cutoff? What about people who report "some difficulty" to more than one functional area? What about those who only report "some difficulty" to one area? And what is the difference between those who overlap or don't overlap between EDM and the WG-SS using different thresholds? Disability isn't really a binary variable -- functional difficulties exist along a continuum. What is gained from the WG-SS compared to the EDM because of the more detailed data?
Exploring these things would add significantly to the paper.
There is often a mistaken belief that there is a "gold standard" as to what constitutes a disability. Different measures capture different aspects and it is important to understand them. The good thing about the WG-SS questions is that they were extensively cognitively tested so we have a sense of how people are responding. Is the same true for the EDM?
